# Prognostic Significance of β-Catenin in Relation to the Tumor Immune Microenvironment in Oral Cancer

**DOI:** 10.3390/biomedicines11102675

**Published:** 2023-09-29

**Authors:** Paloma Lequerica-Fernández, Tania Rodríguez-Santamarta, Eduardo García-García, Verónica Blanco-Lorenzo, Héctor E. Torres-Rivas, Juan P. Rodrigo, Faustino J. Suárez-Sánchez, Juana M. García-Pedrero, Juan Carlos De Vicente

**Affiliations:** 1Department of Biochemistry, Hospital Universitario Central de Asturias (HUCA), Carretera de Rubín, 33011 Oviedo, Spain; palomalequerica@gmail.com; 2Instituto de Investigación Sanitaria del Principado de Asturias (ISPA), Instituto Universitario de Oncología del Principado de Asturias (IUOPA), Universidad de Oviedo, Carretera de Rubín, 33011 Oviedo, Spain; taniasantamarta@gmail.com (T.R.-S.); jprodrigo@uniovi.es (J.P.R.); 3Department of Oral and Maxillofacial Surgery, Hospital Universitario Central de Asturias (HUCA), Carretera de Rubín, 33011 Oviedo, Spain; edu.gargar.95@gmail.com; 4Department of Pathology, Hospital Universitario Central de Asturias (HUCA), Carretera de Rubín, 33011 Oviedo, Spain; veronica.blanco@sespa.es (V.B.-L.); ress_444@yahoo.com (H.E.T.-R.); 5Department of Otolaryngology, Hospital Universitario Central de Asturias (HUCA), Carretera de Rubín, 33011 Oviedo, Spain; 6Department of Surgery, University of Oviedo, 33011 Oviedo, Spain; 7Centro de Investigación Biomédica en Red de Cáncer (CIBERONC), Instituto de Salud Carlos III, Av. Monforte de Lemos, 28029 Madrid, Spain; 8Department of Pathology, Hospital Universitario de Cabueñes, Prados, 33394 Gijon, Spain; faustinosuarezsanchez@gmail.com

**Keywords:** oral squamous cell carcinoma, β-catenin, PD-L1, CD8^+^ lymphocytes, tumor microenvironment, prognosis

## Abstract

The aim of this study was to investigate the prognostic relevance of β-catenin expression in oral squamous cell carcinoma (OSCC) and to explore relationships with the tumor immune microenvironment. Expression of β-catenin and PD-L1, as well as lymphocyte and macrophage densities, were evaluated by immunohistochemistry in 125 OSCC patient specimens. Membranous β-catenin expression was detected in 102 (81.6%) and nuclear β-catenin in 2 (1.6%) tumors. There was an association between β-catenin expression, tumoral, and stromal CD8^+^ T-cell infiltration (TIL) and also the type of tumor immune microenvironment (TIME). Tumors harboring nuclear β-catenin were associated with a type II TIME (i.e., immune ignorance defined by a negative PD-L1 expression and low CD8^+^ TIL density), whereas tumors with membranous β-catenin expression were predominantly type IV (i.e., immune tolerance defined by negative PD-L1 and high CD8^+^ TIL density). Combined, but not individual, high stromal CD8^+^ TILs and membranous β-catenin expression was independently associated with better disease-specific survival (HR = 0.48, *p* = 0.019). Taken together, a combination of high stromal CD8^+^ T-cell infiltration and membranous β-catenin in the tumor emerges as an independent predictor of better survival in OSCC patients.

## 1. Introduction

Oral squamous cell carcinoma (OSCC) represents more than 90% of all oral malignancies, with over 377,713 new cases reported worldwide in the year 2020. It is characterized by poor prognosis, with a five-year mortality rate close to 50% [1], despite recent advances in therapy. It is therefore necessary to possess a deeper knowledge of tumor biology in order to improve OSCC treatment and patient survival. Immunotherapy is commonly used nowadays for the treatment of many cancers, and the PD1/PD-L1 axis has emerged in recent years as a key complex to maintain a balance between immune tolerance and immunopathology. Programmed cell death protein 1 (PD-1) is an inhibitory immune checkpoint expressed on the surface of T cells [2]. PD-L1 acts as main ligand of the PD-1 receptor, and it is expressed in activated T cells, B cells, dendritic cells, macrophages, and certain tumor types, including OSCC [2]. Tumor PD-L1 binding to PD-1 on T cells inhibits the CD8^+^ T-cell activation/functions and promotes the induction of regulatory or suppressor T cells (Tregs), contributing to cancer immune evasion [3,4]. In turn, anti-PD-1/PD-L1 treatments block the interaction between PD-1 and its ligand, thereby restoring T-cell activity and the anti-tumor immune response. However, around 66–85% of patients do not respond to immunotherapy or show any significant clinical benefit [5]. 

Tumors interact continually with the surrounding microenvironment (TME), composed of diverse immune cells, fibroblasts, blood vessels, signaling molecules, and the extracellular matrix [6]. A lymphocytic reaction characterized by high T-cell infiltration (defined as a T cell-inflamed TME) is commonly associated with a favorable clinical outcome in OSCC patients, thus supporting the importance of T cell-mediated immunity in tumor clearance [7]. Nevertheless, it still remains unclear whether the TME plays a direct role in PD-L1 transcription regulation and tumor immune evasion [8]. Recently, a TCGA database study demonstrated that most cancers are inversely associated with a T cell-inflamed gene expression signature, hence emerging Wnt/β-catenin signaling activation as a potential causal pathway [9]. 

The Wnt/β-catenin signaling pathway encompasses two major categories: the canonical and the non-canonical Wnt pathway. The key elements involved in the canonical Wnt/β-catenin signaling pathway include Wnt, Frizzled (Fz) receptors, low-density lipoprotein-related protein 5/6 (LRP5/6) co-receptors, destruction complex components [adenomatous polyposis coli protein (APC), Axin, glycogen synthetase 3 (GSK3β), and casein protein kinase 1α (CK1 α)], Dishevelled (DVL), β-catenin, and T-cell factor (TCF)/lymphoid enhancer factor (LEF) transcription factors [10]. When Fz receptors are unoccupied, cytoplasmic β-catenin is degraded by the destruction complex after its sequential phosphorylation and subsequent ubiquitylation [11]. The first phosphorylation is at Ser45 by CK1α, and then at Thr41, Ser37, and Ser 33 by GSK3β. However, when the Wnt ligand binds to the Fz receptor and its LRP5/LRP6 co-receptor, Wnt signaling is activated [10], which results in Axin translocation and DVL phosphorylation, and ultimately destruction complex dissociation [12]. Subsequently, Axin, GSK3β, and CK1 migrate from the cytoplasm to the cell membrane, thus resulting in β-catenin stabilization through dephosphorylation. Stable β-catenin translocates into the cell nucleus where it interacts with TCF/LEF transcription factors to induce the expression of Wnt target genes, such as Axin-2, c-Myc, cyclin D1, ITF-2, Lgr5, MMP-1, MMP-7, and PPAR-δ [11,13], thereby altering cellular processes such as proliferation, differentiation, and stemness, and promoting cancer cell proliferation and survival [4,10]. In addition, there are two non-canonical Wnt pathway categories: the Wnt/PCP (planar cell polarity) that participates in cell polarity and migration and the Wnt/Ca^2+^ pathway that is crucial for cell adhesion and motility during gastrulation [4].

The Wnt/β-catenin pathway is aberrantly activated in numerous tumor types [4], including OSCC [14]. The altered expression of β-catenin correlates with oral tumorigenesis, and it has also been linked to a poor prognosis in OSCC [15]. Wnt/β-catenin has been considered the most important pathway in OSCC [16]. In a recent meta-analysis including 41 studies and 2746 OSCC patients, the loss of membrane expression, cytoplasmic expression, and/or nuclear expression of β-catenin was found a poor prognostic factor [17]. Furthermore, a novel role of β-catenin has been described, regulating PD-L1 transcription [8]. β-catenin activation increased PD-L1 transcription, limited CD8^+^ T-cell activation, and promoted tumor growth. Conversely, β-catenin depletion reduced PD-L1 expression levels in tumor cells, enhanced CD8^+^ T-cell infiltration, and inhibited tumor growth [8].

In this study, we investigated the significance of the β-catenin expression pattern in a large cohort of OSCC patients using tissue microarray immunohistochemistry, its association with the type of tumor immune microenvironment (TIME), and the potential prognostic relevance in OSCC.

## 2. Materials and Methods

### 2.1. Patients and Tissue Specimens

A cohort of 125 OSCC patients who received surgical treatment at the Hospital Universitario Central de Asturias (HUCA) between 1996 and 2007 was retrospectively selected. This study was conducted following the ethical criteria of the Declaration of Helsinki and approved by the HUCA Ethics Committee and also by the Regional CEIC from Principado de Asturias (date of approval 14 May 2019; approval number 136/19, for the project PI19/01255). Written informed consent was obtained from all patients. We retrieved clinical and histopathologic information from the patients’ files, and pathology reports, which are summarized in Table 1. The clinical staging was determined according to the 8th edition of the AJCC TNM classification [18] and the histological grading according to the WHO classification [19]. 

Formalin-fixed paraffin-embedded (FFPE) tissue samples were sourced from the Principado de Asturias BioBank (PT20/0161) and processed following standard operating procedures. Clinicopathologic data were collected from clinical records. The inclusion criteria for all the participants enrolled were: (i) primary OSCC (International Classification of Disease-10 diagnosis codes: C02.0, C02.1, C02.2, C02.3, C03.0, C03.1, C04.0, C04.1, C05.0, C06.0, C06.1, and C06.2), (ii) treatment-naïve patients from whom we had formalin-fixed paraffin-embedded (FFPE) primary biopsy tissue samples, and (iii) with a minimum follow-up of at least three years for alive patients. In addition, the exclusion criteria were: (i) OSCC with immediate postoperative death, (ii) recurrent disease, (iii) neoadjuvant chemo- or radiotherapy, and (iv) missing survival data. 

All patients underwent surgery of the primary tumor with curative intention as well as neck dissection. None of the patients received any treatment before surgery, but complementary radiotherapy and/or chemotherapy were administered in 75 (60%) and 14 (11%) cases, respectively. During the follow-up period (6 to 230 months), 19 (15%) patients suffered from a second primary tumor in the oral cavity, and 51 (42%) patients died of OSCC. The clinical endpoint of this study was disease-specific survival (DSS), calculated as the period of time between the initial treatment and the death caused by the tumor or the presence of a non-treatable recurrence.

Samples and data from donors included in this study were provided by the Principado de Asturias BioBank (PT20/0161), integrated in the Spanish National Biobanks and Biomodels Network financed with European funds and they were processed following standard operating procedures with the appropriate approval of the Ethical and Scientific Committees.

### 2.2. Immunohistochemistry (IHC)

Tissue microarrays (TMAs) were constructed by collecting 1 mm tissue cores from the most morphologically representative areas of formalin-fixed, paraffin-embedded (FFPE) tissue blocks. Three individual cores were taken per tumor block. Then, 3 μm tissue sections dried on Flex IHC microscope slides (DakoCytomation, Glostrup, Denmark) were heated with high-pH Envision Flex Target Retrieval solution (Dako) and stained on an automatic staining workstation (Dako Autostainer Plus, Dako) using monoclonal antibodies against β-catenin (BD Biosciences, #610153, 1:200 dilution), CD8 (Dako, clone C8/144B, prediluted), FoxP3 (Cell Signaling Technology, Danvers, MA, USA, clone D6O8R, 1:100 dilution), PD-L1 antibody (PD-L1 IHC 22C3 pharmDx, Dako SK006, clone 22C3, 1:200 dilution), CD20 (Dako, clone L26, #M0755; 1:200 dilution), CD4 (Dako, clone 4B12, 1:80 dilution), CD68 (Agilent-Dako, Santa Clara, CA, USA, clone KP1, prediluted), and CD163 (Biocare Medical, Pacheco, CA, USA, clone 10D6, 1:100 dilution). The antibody-antigen complexes were visualized using the Dako EnVision Flex + Visualization System (Dako) and diaminobenzidine chromogen as a substrate. 

Each TMA also contained three cores of morphologically normal oral mucosa from non-oncological patients undergoing oral surgery, used as internal controls. No staining was observed when the primary antibody was omitted. Negative control was included by replacing the primary antibody with serum and positive controls using appropriate positive control tissue slides. 

The IHC results were independently evaluated by two observers (VB-L and FJS-S), blinded to clinical information. The number of CD20^+^, CD68^+^, CD163^+^, CD4^+^, CD8^+^, and FOXP3^+^ cells in both the tumor nests and the tumor stroma was counted in each 1 mm^2^ area from three independent HPFs at 400x, as we previously reported [20,21,22]. The median value was used as a cut-off point to separate patients based on the tumoral and stromal CD8^+^ lymphocyte densities into two groups, above (high density) and below (low density) the median number of positive stained cells for the total patient population. Additionally, TIL intensity in the tumor nests and the surrounding stroma was subdivided into three groups: negative, mild-moderate, or intense. PD-L1 expression in more than 10% of the tumor cells was previously found to significantly associate with poorer survival [23], and therefore established as a cut-off point for subsequent analyses. Since β-catenin staining intensity within the tumor showed a homogeneous pattern, a semiquantitative scoring system based on staining positivity was applied for IHC evaluation into three categories: negative (0), membrane staining, and (1) nuclear staining (2). The type of tumor immune microenvironment (TIME) based on the presence of TILs and PD-L1 expression was determined according to the classification reported by Teng et al. [24].

### 2.3. Statistical Analysis

Statistical analyses were carried out using SPSS software version 27 (IBM Co., Armonk, NY, USA). Continuous variables were expressed as the mean ± standard deviation (SD), and absolute and relative frequencies were calculated for categorical variables. Associations between the numbers of immune cells infiltrating the tumors and β-catenin expression were assessed by using the Kruskall–Wallis test. Fisher’s exact test was used to evaluate the relationship between β-catenin expression, the type of TIME, and TIL intensity. Potential associations of β-catenin and the type of TIME with the different clinicopathological variables were assessed using the Chi-square or Fisher’s exact test. 

Disease-specific survival (DSS) was estimated using the Kaplan–Meier method, and the log-rank test was used for comparisons between survival rates. Hazard ratios (HR) with their 95% confidence intervals (CI) were calculated using univariable and multivariable Cox regression models. All tests were two-sided, and *p*-values less than 0.05 were considered statistically significant. 

## 3. Results

### 3.1. Immunohistochemical Analysis of β-Catenin, PD-L1, and CD8^+^ TIL Density in OSCC Patient Samples

β-catenin immunostaining was evaluated in 125 OSCC samples. Membrane β-catenin expression was detected in 102 (81.6%) tumors, nuclear β-catenin was only found in 2 cases (1.6%), and no expression of β-catenin neither in the membrane nor the nucleus was observed in 21 cases (16.8%). Representative images of positive β-catenin staining (membranous and nuclear) are shown in Figure 1. A total of 104 cases (85%) exhibited positive PD-L1 immunostaining in tumor cells. The mean CD8^+^ TILs in the tumor nests was 47.88 ± 57.16 cells per mm^2^ (range: 0.00 to 288.33), and the mean stromal CD8^+^ TILs 178.45 ± 203.21 (range: 0.33 to 1202.67) cells per mm^2^. 

### 3.2. Associations between the Expression of β-Catenin, PD-L1, and CD8^+^ TIL Density in OSCC

There was an association between the infiltration of CD8^+^ T cells in both stroma and tumor nests, and β-catenin immunoexpression (Table 2). The mean number of stromal CD8^+^ T cells was higher in tumors harboring negative β-catenin expression, and the lowest CD8^+^ TIL density was found in those tumors with positive nuclear β-catenin (Kruskal–Wallis test, *p* = 0.02). Similarly, the mean number of tumoral CD8^+^ T cells was higher in tumors with negative β-catenin expression, and also the lowest CD8^+^ TIL density was seen in the subset of tumors with positive nuclear β-catenin (Kruskal–Wallis test, *p* = 0.029). Nevertheless, since only two cases exhibited nuclear β-catenin, these figures should be cautiously taken into account. Moreover, there was a tendency of association between PD-L1 positivity and membranous β-catenin expression, although this relationship did not reach statistical significance (Fisher’s exact test, *p* = 0.25).

We also assessed the association between β-catenin expression and the type of TIME, which had a statistically significant relationship (Fisher’s exact test, *p* = 0.012). Tumors harboring nuclear β-catenin were associated with type II TIME (i.e., immunological ignorance), whereas membranous β-catenin was predominantly associated with type IV (i.e., immune tolerance) (Table 3). 

We next assessed the correlation between different immune cell subtypes and the density of stromal and tumoral TIL infiltration, and we found significant associations between intense TIL infiltration and tumoral CD68^+^ (*p* < 0.0001), CD163^+^ (*p* = 0.006), and tumoral CD8^+^ (*p* = 0.02) (Table 4).

### 3.3. Associations with Clinicopathological Features and Patient Survival

Absence/loss of β-catenin expression or nuclear β-catenin expression were more frequently observed in larger tumor sizes (T3–T4) (21% vs. 17% in T1-T2), tumors with neck lymph node metastasis (pN+) (25% vs. 15% in pN0 cases), and recurrent tumors (22% vs. 16% in non-recurrent tumors); however, these differences did not reach statistical significance. Regarding histological degree of differentiation, absence or nuclear β-catenin expression was more frequently detected in moderately and poorly differentiated tumors, (22% vs. 16% in well-differentiated tumors), almost reaching significance (*p* = 0.06).

Stromal/tumoral CD8^+^ TILs were divided into two categories (high vs. low density) according to their respective median values. Only stromal CD8^+^ TILs, but not tumoral CD8^+^ TILs, were significantly associated with DSS (0.8 and 0.73 patient survival in high vs. low TIL infiltration, *p* = 0.045) (Figure 2A) as previously reported [20]. Regarding β-catenin expression, membrane staining was associated with better survival than nuclear β-catenin or no expression, but differences were not statistically significant (Figure 2B). However, when combining the stromal CD8^+^ TILs with membranous β-catenin expression in the tumor, we found that patients harboring high CD8^+^ TIL density (above the median value) and positive membranous β-catenin showed the highest DSS (*p* = 0.012) (Figure 2C).

Multivariable Cox regression analysis, including T classification (T1-T2 vs. T3-T4), N classification (N0 vs. N+), differentiation grade (well-moderately vs. poorly differentiated), and stromal CD8^+^ TIL infiltration combined with tumoral β-catenin expression (high CD8^+^ TIL density and positive membranous β-catenin vs. all the rest of the cases) showed that the parameters independently associated with worse DSS were T3-T4 classification (HR = 2.61, 95% CI = 1.45–4.68, *p* = 0.001), and neck node metastasis (HR = 1.84, 95% CI = 1.06–3.18, *p* = 0.02). Conversely, high CD8^+^ TIL density combined with membranous β-catenin was a significant independent predictor of better DSS (HR = 0.48, 95% CI = 0.26–0.88, *p* = 0.019). Finally, the degree of differentiation was not significantly associated with better DSS (HR = 0.56, 95% CI = 0.31–1.03, *p* = 0.06). If we included in the multivariable analysis the stromal CD8^+^ TIL infiltration instead of this variable combined with tumoral β-catenin expression, T and N parameters retained their significant association with a worse DSS; however, CD8^+^ TILs density did not reach a significant independent association with survival (HR = 0.60, 95% CI = 0.34–1.06, *p* = 0.08).

## 4. Discussion

β-catenin is an 88 kDa multifunctional and evolutionary-conserved protein, a member of the armadillo family of proteins, and involved in two independent processes, which are cell-cell adhesion and signal transduction [25]. β-catenin is considered as the ‘gatekeeper’ of the canonical Wnt signaling [26]. In the absence of Wnt ligands, this protein is involved in cell adhesion, thereby acting as a bridge between E-cadherin and cytoskeleton-associated actin to form adherent junctions between adjacent cells in the stratified squamous epithelium of oral mucosa [27]. In this series, expression of membranous β-catenin was detected in 81.6% tumors, nuclear β-catenin was only found in two cases (1.6%), and β-catenin expression was absent in 16.8%. Akinyamoju et al. [28] reported positive β-catenin expression in 70.7% OSCC samples. Laxmidevi et al. [29] reported 56.6% of β-catenin positivity and Zaid [30] found 67.1% in OSCC.

Herein, we did not find any significant relationship between β-catenin expression and the clinicopathological characteristics of our sample. Nevertheless, abnormal β-catenin expression (defined as absence/loss of expression or nuclear expression) was more frequent in T3-T4 tumors, with the presence of nodal metastasis as well as recurrent OSCC tumors. Similar to our results, Al-Rawi et al. [31] found no significant associations with either clinicopathological features or prognosis. In a study with 30 OSCC samples, β-catenin expression loss evaluated by immunohistochemistry was an unreliable marker of nodal metastasis; however, the loss of this protein was associated with a lower degree of differentiation [32]. Conversely, Tanaka et al. [33] reported a significant reduction in the expression level of β-catenin in those tumors with lymph node metastasis compared with non-metastatic cases. Interestingly, we found a trend between β-catenin expression loss and poor tumor differentiation. Concordantly, Zaid [30] and Zhao et al. [34] showed a significant reduction in β-catenin expression in poor histopathological grades in OSCC and in esophageal squamous cell carcinoma, respectively. Since only two cases showed nuclear β-catenin staining in our OSCC cohort, consequently, these results should be cautiously interpreted. Some studies reported that reduced nuclear β-catenin is related with a more aggressive tumor behavior in non-small cell lung cancer [35], whereas Pukkila et al. [36] did not find such an association in squamous cell carcinomas of the oropharynx and hypopharynx. Furthermore, nuclear β-catenin expression was associated with shorter survival rates. Moreover, in a meta-analysis that included 2746 OSCC patients, Ramos-García and Gonzalez-Moles [17] showed that aberrant β-catenin expression was significantly associated with poor overall survival, disease-free survival, higher tumor size, neck lymph node metastasis, and moderately-poorly differentiated tumors.

Among the reasons that may explain the differences and inconsistencies from the available data are small sample sizes in most studies, methodological differences in immunohistochemical evaluation and scoring, often subjective, and different antibodies/epitopes used. 

In this study, we showed that β-catenin expression in the nucleus of oral cancer cells was associated with the lowest density of CD8^+^ TILs in the TME, both in the tumor as well as in the surrounding stroma, and with a tendency to a low tumoral PD-L1 expression. The interaction between PD-1 and PD-L1 inhibits the activation and effector functions of CD8^+^ T cells and induces suppressive Treg cells, ultimately leading to cancer immune evasion [8]. Oncogenic pathways may contribute to the protumoral effects of cancer-intrinsic PD-L1. Specifically, the Wnt/β-catenin signaling pathway could exert a relevant role in facilitating PD-L1-mediated cell proliferation, migration, and invasion. PD-L1 expression induces ERK phosphorylation and activates the Wnt/β-catenin signaling pathway, which upregulates downstream target genes, as reported in colorectal carcinoma and in non-small cell lung cancer [37,38]. In turn, activation of β-catenin increases PD-L1 transcription and promotes immune evasion [8]. According to these data, PD-L1 and β-catenin are reciprocally implicated in a bidirectional positive feedback loop. Therefore, blocking Wnt/β-catenin signaling could increase anti-PD-L1 treatment efficacy [37]. 

Tumors are closely related to the TME, and there is a continuous interplay between tumor cells and stromal cells. It has been recently reported that most tumors are inversely related to a T cell-inflamed gene expression signature, potentially linked to Wnt/β-catenin pathway activation [9]. Here, we found a relationship between tumoral and stromal CD8^+^ TIL density and β-catenin expression in OSCC. Furthermore, tumors harboring nuclear β-catenin were associated with a type II TIME (i.e., immunological ignorance), as reported in other HNSCC subsites [39]. Notably, a T cell-inflamed phenotype is associated with the efficacy of an immune checkpoint blockade, whereas non-T-cell inflamed tumors rarely benefit from anti-PD-1 therapies [2,39]. We also found an association between the nuclear β-catenin and poor survival, in accordance with previous reports [40]. The molecular mechanism by which tumoral β-catenin regulates PD-L1-mediated immunosuppression remains unknown [41]. However, it has been recently shown that β-catenin can be activated by both Wnt ligand and epidermal growth factor receptor (EGFR) activation (frequent in OSCC), which leads to the binding of the β-catenin/TCF/LEF complex to the promoter region of the *CD274* gene to induce PD-L1 expression [8]. 

The Wnt/β-catenin pathway has been identified as one of the most important oncogenic signaling pathways associated with tumor immune evasion [42], and its mutation leads to a non-T cell inflammatory tumor phenotype. The Wnt/β-catenin signaling pathway also regulates the differentiation of CD4^+^ helper T cells [2] and limits the immunosuppressive activity of Treg cells by modulating FOXP3 transcriptional activity, which has been associated with a lack of T-cell infiltration in the TME of metastatic melanoma and other cancer types [43]. Additionally, the Wnt/β-catenin pathway has also been implicated in the regulation of innate immunity. Colon cancer cells induce IL-β in macrophages via Snail, which is a product of Wnt target genes [44]. Moreover, macrophages usually express higher PD-L1 levels compared with tumor cells [45]. 

Furthermore, we also found a significant relationship between the density of TILs within the tumor nests and stroma, with the density of CD68^+^ and CD163^+^ macrophages infiltrating the tumor, and also with the tumor-infiltrating CD8^+^ T cells. In a recent review, Li et al. [2] summarized the effects of Wnt/β-catenin signaling on cancer immunosurveillance. Briefly, Wnt ligands released by cancer cells induce the canonical Wnt signaling pathway, whose hallmark is the accumulation of β-catenin protein into the nucleus, thereby leading to the inhibition of CD8^+^ T-cell infiltration within the TME and production of Treg cells with subsequent inhibition of cytotoxic T-cell activity. Glycogen synthase kinase 3 (GSK3) inhibition in cancer cells promotes β-catenin activation and subsequently PD-L1 stabilization, driving in turn cytotoxic T-cell activity exhaustion. On the other hand, non-canonical Wnt signaling is initiated by Wnt5a-type ligands [11]. Wnt1 ligands related with the canonical Wnt pathway promote invasion and inhibit apoptosis in OSCC [46], whereas Wnt5a, Wnt5b, Wnt7a, and Wnt7b ligands, all of them related to the non-canonical Wnt pathway, enhance cell migration and invasion in OSCC [47,48,49,50]. The relationship between β-catenin and a non-T-cell inflamed microenvironment has been previously reported by Luke et al. [9]. In an integrative TCGA analysis by separating tumors according to their T-cell inflamed status, they found that 33.9% of tumors were non-T-cell-inflamed, which can be regulated by oncogenic events [9,51]. Furthermore, it was also revealed that over 90% of tumor types across the TCGA showed an inverse correlation between Wnt/β-catenin pathway activation and a T-cell-inflamed gene expression signature.

As far as we know, this is the first study to analyze the relationship between β-catenin expression and the immune infiltrate TME in OSCC, mainly focused on CD8^+^ T cells. 

## 5. Conclusions

In conclusion, our findings support that the Wnt/β-catenin pathway could be a molecular target of paramount relevance in an effort to elicit an infiltration of CD8^+^ T cells in the OSCC TME and potentially increase the efficacy of immunotherapy in the clinical setting. Hence, β-catenin emerges as a valuable molecular target to improve the efficacy of immunotherapy in OSCC by increasing TIL density in the TME. Notably, a combination of stromal CD8^+^ T-cell infiltration and membranous β-catenin expression in the tumor was found to be an independent predictor of better DSS for OSCC patients.

## Figures and Tables

**Figure 1 biomedicines-11-02675-f001:**
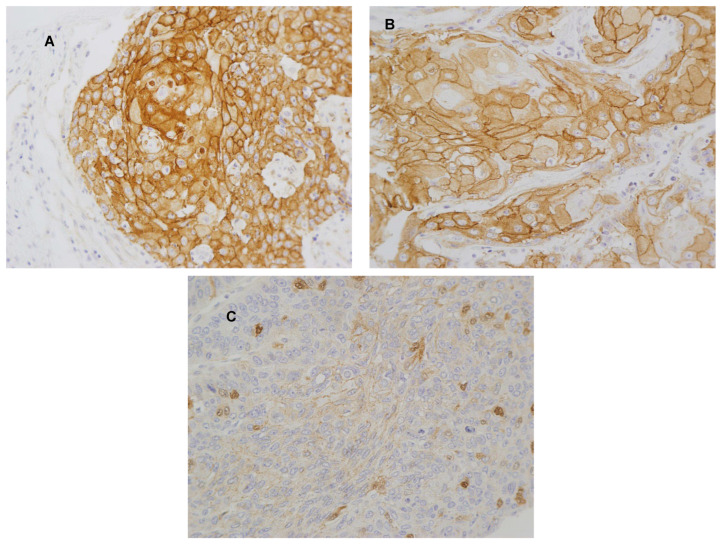
Immunohistochemical analysis of β-catenin in OSCC tissue specimens. Representative images of tumors showing (**A**) membrane and nuclear staining, (**B**) membrane staining, and (**C**) nuclear staining. Magnification 40×.

**Figure 2 biomedicines-11-02675-f002:**
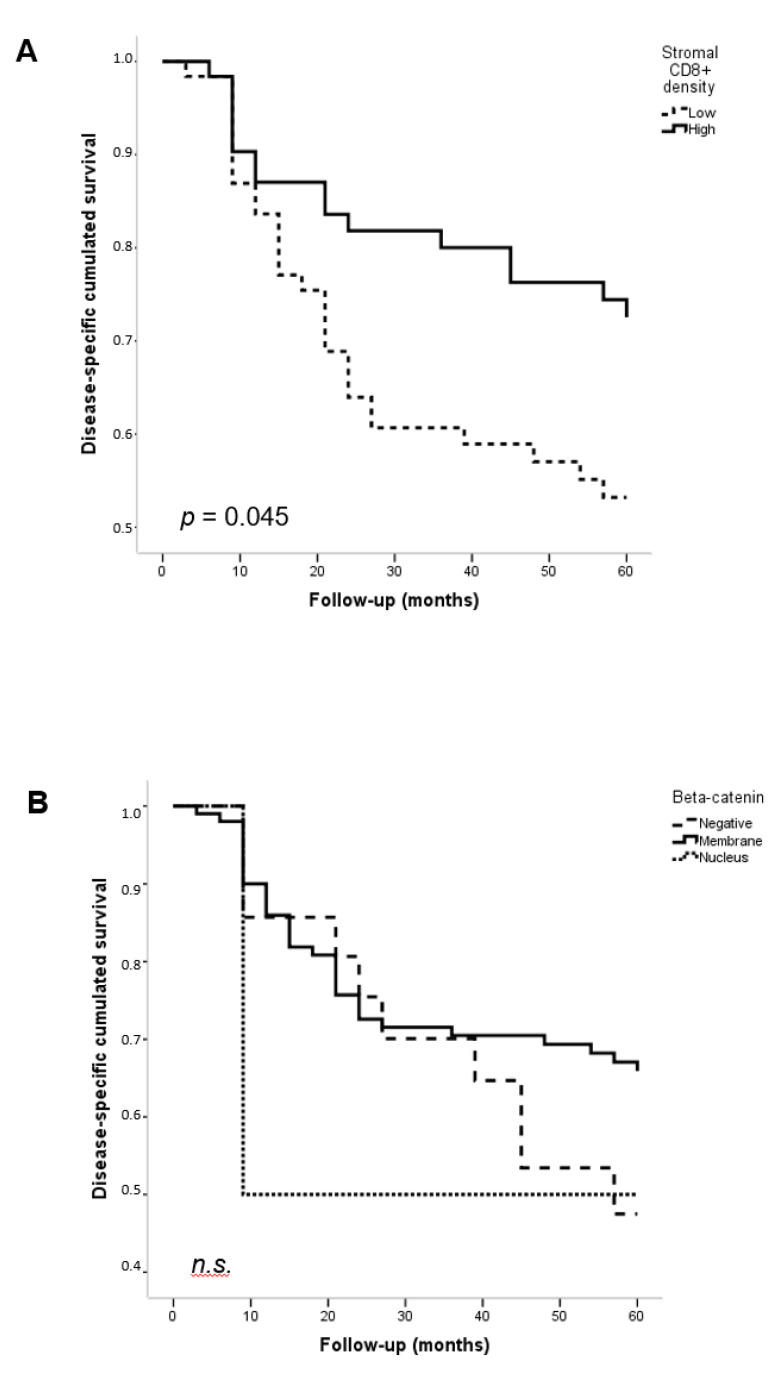
Kaplan–Meier disease-specific survival and overall survival curves in the cohort of 125 OSCC patients categorized by (**A**) High versus low stromal CD8^+^ density (*p* = 0.045), (**B**) β-catenin immunostaining (membrane, nucleus, negative) (non-significant), and (**C**) High CD8^+^ TILs stromal infiltration and membrane β-catenin immunostaining versus rest of cases (*p* = 0.019). *p* values estimated by log-rank test.

**Table 1 biomedicines-11-02675-t001:** Clinical and pathological characteristics of 125 patients with oral squamous cell carcinoma.

Variable	Number (%)
Age (year) (mean ± SD; median; range)	58.69 ± 14.34; 57; 28–91
Gender	
Men	82 (66)
Women	43 (34)
Tobacco use	
Smoker	84 (67)
Non-smoker	41 (33)
Alcohol use	
Drinker	69 (55)
Non-drinker	56 (45)
Location of oral squamous oral cell carcinoma	
Tongue	51 (41)
Floor of the mouth	37 (30)
Gum	22 (18)
Buccal	7 (5)
Retromolar	6 (5)
Palate	2 (1)
Tumor status	
pT1	27 (22)
pT2	54 (43)
pT3	16 (13)
pT4	28 (22)
Nodal status	
pN0	76 (61)
pN1	25 (20)
pN2	24 (19)
Clinical stage	
Stage I	20 (16)
Stage II	32 (26)
Stage III	26 (21)
Stage IV	47 (37)
G status	
G1	80 (64)
G2	41 (33)
G3	4 (3)

**Table 2 biomedicines-11-02675-t002:** Associations between stromal and tumoral CD8^+^ TIL density and β-catenin expression in OSCC patients.

CD8^+^ Compartment	β-Catenin Expression	No. Cases	Mean CD8^+^ TIL Density (SD)	*p*
**Stromal**	Negative Membrane Nucleus	21 102 2	280.15 (279.15) 160.43 (179.83) 30.00 (41.48)	0.020
**Tumoral**	Negative Membrane Nucleus	21 102 2	67.41 (54.74) 44.63 (57.40) 6.83 (3.50)	0.029

SD: standard deviation. *p*-values were calculated using the Kruskal–Wallis test.

**Table 3 biomedicines-11-02675-t003:** Associations between β-catenin expression and the type of immune tumor microenvironment (TME).

Type of Immune TME	β-Catenin Expression			*p*
	Negative	Membrane	Nucleus	
**Type I (PD-L1+/CD8** ^ **+** ^ ** high)**	3 (16%)	10 (10%)	0 (0%)	
**Type II (PD-L1−/CD8** ^ **+** ^ ** low)**	1 (5%)	37 (37%)	2 (100%)	0.012
**Type III (PD-L1+/CD8** ^ **+** ^ ** low)**	2 (11%)	3 (3%)	0 (0%)	
**Type IV (PD-L1−/CD8** ^ **+** ^ ** high)**	13 (68%)	51 (50%)	0 (0%)	

*p*-value calculated using Fisher’s exact test.

**Table 4 biomedicines-11-02675-t004:** Correlations between the mean numbers of CD68^+^ and CD163^+^ macrophages and CD8^+^, CD20^+^, CD4^+^, and FOXP3^+^ infiltrating TILs in the tumor nests and surrounding stroma, according to the intensity of TIL infiltration.

Mean (SD)	TILs			*p*
	Negative	Mild-Moderate	Intense	
**Stromal CD68** ^ **+** ^	110.55 (73.53)	140.56 (94.67)	118.72 (54.29)	0.20
**Tumoral CD68** ^ **+** ^	37.86 (36.65)	61.39 (47.64)	115.66 (36.22)	<0.0001
**Stromal CD163** ^ **+** ^	165.35 (90.93)	172.52 (109.10)	149.66 (81.73)	0.92
**Tumoral CD163** ^ **+** ^	25.79 (27.58)	36.19 (28.62)	54.33 (32.04)	0.006
**Stromal CD8** ^ **+** ^	168.86 (203.92)	202.73 (211.62)	83.66 (100.49)	0.11
**Tumoral CD8** ^ **+** ^	34.65 (46.44)	62.46 (64.67)	76.27 (70.72)	0.02
**Stromal CD20** ^ **+** ^	36.15 (73.23)	54.48 (88.84)	17.50 (35.96)	0.06
**Tumoral CD20** ^ **+** ^	1.21 (2.98)	2.34 (3.87)	0.50 (0.91)	0.26
**Stromal CD4** ^ **+** ^	52.81 (78.38)	61.48 (56.03)	19.27 (8.58)	0.08
**Tumoral CD4** ^ **+** ^	4.66 (10.40)	8.43 (14.83)	1.38 (1.28)	0.05
**Stromal FOXP3** ^ **+** ^	12.62 (18.38)	19.65 (31.70)	17.16 (15.30)	0.51
**Tumoral FOXP3** ^ **+** ^	2.10 (3.66)	4.30 (9.07)	4.83 (4.54)	0.20

The Kruskal–Wallis *p* values are shown.

## Data Availability

Data supporting the present study are available from the corresponding author (JCV) upon reasonable request.

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
