# Peer review of "Prognostic Significance of β-Catenin in Relation to the Tumor Immune Microenvironment in Oral Cancer"

_biomedicines, 2023, doi:10.3390/biomedicines11102675_

Round 1
Reviewer 1 Report
Comments
In this manuscript, the authors investigated the protein expression of beta-catenin in oral squamous cell carcinoma (OSCC), and assessed its correlation with the infiltrating immune cells in the tumor microenvironment (TME). Wnt/beta-catenin signaling pathway plays pivotal roles in several types of malignancies, and some reports suggested the involvement of this signaling pathway in the TME regulation.
The data demonstrated that beta-catenin protein expression was seen in the majority of the cases studied, and its location was observed both in membrane and nucleus. The combined data of immunohistochemistry have shown that stromal and tumoral CD8+ cells were significantly decreased in the cases with membranous beta-catenin expression, which may influence the sensitivity of immune checkpoint inhibitors.
The data seemed interesting. However, because of a severe defect, this manuscript is insufficient for publication. The points are as follows;
First, the data showed that the majority of the OSCC cases exhibited membranous beta-catenin expression, which did not affect significant influence on disease-specific survival. Indeed, CD8+ immune cell status does affect the prognosis, no direct evidence that membranous beta-catenin expression promotes the particular immune cell infiltration was shown.
Second, the meaning of beta-catenin expression in OSCC cases, i.e. correlation between beta-catenin expression and clinical parameters, was not discussed.
Author Response
REVIEWER # 1
Comments and Suggestions for Authors
In this manuscript, the authors investigated the protein expression of beta-catenin in oral squamous cell carcinoma (OSCC), and assessed its correlation with the infiltrating immune cells in the tumor microenvironment (TME). Wnt/beta-catenin signaling pathway plays pivotal roles in several types of malignancies, and some reports suggested the involvement of this signaling pathway in the TME regulation.
The data demonstrated that beta-catenin protein expression was seen in the majority of the cases studied, and its location was observed both in membrane and nucleus. The combined data of immunohistochemistry have shown that stromal and tumoral CD8+ cells were significantly decreased in the cases with membranous beta-catenin expression, which may influence the sensitivity of immune checkpoint inhibitors.
Response: We thank the reviewer for minutely going through the results and findings of our study.
The data seemed interesting. However, because of a severe defect, this manuscript is insufficient for publication. The points are as follows;
Response: Many thanks for considering that the data presented are interesting and for the insightful recommendations to improve the scientific content of our manuscript.
First, the data showed that the majority of the OSCC cases exhibited membranous beta-catenin expression, which did not affect significant influence on disease-specific survival. Indeed, CD8+ immune cell status does affect the prognosis, no direct evidence that membranous beta-catenin expression promotes the particular immune cell infiltration was shown.
Response: As stated in the abstract and Results section, tumors with membranous β-catenin expression were predominantly type IV (i.e. immune tolerance defined by negative PD-L1 and high CD8+ TIL density). In fact, stromal CD8+ TILs was significantly associated with DSS (p = 0.045; shown in Figure 2A), which is in line to previous reports [20]. More importantly, the combination of stromal CD8+ TIL infiltration and positive membranous β-catenin in the tumor showed the highest DSS (p = 0.012; Figure 2C), and this combination was revealed a significant independent predictor of better DSS (HR = 0.48, 95% CI = 0.26 – 0.88, p = 0.019) in the multivariable Cox analysis. In our OSCC patient cohort, the combination of stromal CD8+ TIL infiltration and positive membranous β-catenin remarkably showed superior predictability than CD8+ TIL infiltration alone.
Second, the meaning of beta-catenin expression in OSCC cases, i.e. correlation between beta-catenin expression and clinical parameters, was not discussed.
Response: Thanks for pointing this out. The significance of beta-catenin in the studied OSCC cohort as well as potential correlations with clinical variables are now discussed, according to the reviewer’s suggestion. In addition, additional data have now been included in Results subsection 3.3 regarding possible associations of beta-catenin expression and clinicopathological parameters.
Reviewer 2 Report
"Prognostic Significance of β-Catenin in Relation to the Tumor 2 Immune Microenvironment in Oral Cancer "
Research into oral squamous cell carcinoma continues to be extremely important, as it is often detected and treated at very advanced stages and, combined with the lack of response to therapy, still has a high mortality rate.
In this study, the authors assessed the prognostic significance of β-Catenin in Relation to the Tumour and Immune Microenvironment in Oral Cancer, which I found interesting, but I don't think it really added much knowledge on the subject. There are other studies already published on the same subject, involving the study of more characteristics, namely HPV+ or HPV - tumours. However, it's another contribution to the subject. The article is well written, with current references involving the main articles on the subject.
In my opinion, I think the article could do with some changes and I'll make a few suggestions:
- The Introduction (Overall)
You must follow the journal template to rewrite the abstract.
You should demonstrate the importance of the study firstly Wnt/β-catenin signalling, the relationship with the other parameters studied and then addressing PD-1/PD-L1. β-catenin is the core of the article.
It should be explored further in the introduction.
- Materials and methods
In line 84 where it says to consult the article to find out the characteristics of the sample - you should describe the characteristics of the sample in the materials and methods, not send it to the article. It makes it difficult to read and understand the sample.
They should improve the description of the immunohistochemical evaluation of the different antibodies.
I wonder if they haven't assessed the intensity of β-catenin
The results
Table 2 is missing the β-catenin legend.
The Discussion
They should keep the same order in the discussion.
Author Response
REVIEWER # 2
Comments and Suggestions for Authors
"Prognostic Significance of β-Catenin in Relation to the Tumor 2 Immune Microenvironment in Oral Cancer"
Research into oral squamous cell carcinoma continues to be extremely important, as it is often detected and treated at very advanced stages and, combined with the lack of response to therapy, still has a high mortality rate.
Response: We thank the reviewer for highlighting that our research tackles an extremely important subject area.
In this study, the authors assessed the prognostic significance of β-Catenin in Relation to the Tumour and Immune Microenvironment in Oral Cancer, which I found interesting, but I don't think it really added much knowledge on the subject. There are other studies already published on the same subject, involving the study of more characteristics, namely HPV+ or HPV - tumours. However, it's another contribution to the subject. The article is well written, with current references involving the main articles on the subject.
Response: We also thank for considering that this work is interesting. Beyond being a validation study in a large independent cohort of European OSCC patients (to date scarcely investigated in this population), and therefore a relevant contribution to the subject, we truly believe that our study also provides some original data with added value to the field. To the best of our knowledge, this is the study to assess the association between β-catenin expression with the type of tumor immune microenvironment (TIME) that uncovered important significant relationships. Thus, tumors harboring nuclear β-catenin were associated with type II TIME (i.e. immunological ignorance), whereas membranous β-catenin was predominantly associated with type IV (i.e. immune tolerance). Interestingly, for a potential clinical application, the combination of stromal CD8+ TIL infiltration and positive membranous β-catenin in the tumor was prognostic, and found to significantly associate with the highest DSS (p = 0.012; Figure 2C). More importantly, this combination was revealed a significant independent predictor of better DSS (HR = 0.48, 95% CI = 0.26 – 0.88, p = 0.019) in the multivariable Cox analysis. In our OSCC patient cohort, the combination of stromal CD8+ TIL infiltration and positive membranous β-catenin remarkably showed superior predictability than CD8+ TIL infiltration alone.
In my opinion, I think the article could do with some changes and I'll make a few suggestions:
Response: Thank you for the positive comments. We appreciate all the valuable recommendations to improve our revised version of the manuscript.
- The Introduction (Overall)
You must follow the journal template to rewrite the abstract.
Response: The abstract has been modified accordingly.
You should demonstrate the importance of the study firstly Wnt/β-catenin signalling, the relationship with the other parameters studied and then addressing PD-1/PD-L1. β-catenin is the core of the article.
It should be explored further in the introduction.
Response: Further details have now been included in the Introduction section to explain the importance of Wnt/beta-catenin signaling as central subject of this study, as suggested.
- Materials and methods
In line 84 where it says to consult the article to find out the characteristics of the sample - you should describe the characteristics of the sample in the materials and methods, not send it to the article. It makes it difficult to read and understand the sample.
Response: Following the reviewer’s suggestion, the characteristics of the studied OSCC cohort are now described in Methods section, as well as summarized in Table 1. We apologize for the missing data in our first version of the manuscript.
They should improve the description of the immunohistochemical evaluation of the different antibodies.
Response: We should clarify that the description of the immunohistochemical evaluation has been modified according to the Editorial indications to avoid overlapping text with other previous works, which have been precisely cited for each protein (Refs. 20-24). Nevertheless, we have maintained a brief summary for each protein, and also full details on the primary antibodies (i.e. catalog numbers, providers) and working dilution used for all of them.
I wonder if they haven't assessed the intensity of β-catenin
Response: β-catenin staining intensity within the tumor showed a homogeneous pattern (this is now precisely indicated in Methods section). For this reason, a semiquantitative scoring system based on staining positivity was applied for IHC evaluation into three categories: negative (0), membrane staining, and (1) nuclear staining (2). Hence, scoring was based on staining positivity in the distinct subcellular locations (i.e. nuclear and membranous), which are relevant for the function of this protein in the context of the Wnt/β-catenin signaling pathway.
The results
Table 2 is missing the β-catenin legend.
Response: Table 2 has now been amended.
The Discussion
They should keep the same order in the discussion.
Response: The Discussion text has been restructured and also considerably extended providing further information, according to the comments from Reviewer 1.
Round 2
Reviewer 1 Report
The authors have appropriately modified the manuscript.